# DLin-MC3-Containing mRNA Lipid Nanoparticles Induce an Antibody Th2-Biased Immune Response Polarization in a Delivery Route-Dependent Manner in Mice

**DOI:** 10.3390/pharmaceutics15031009

**Published:** 2023-03-21

**Authors:** Altan Yavuz, Céline Coiffier, Cynthia Garapon, Serra Gurcan, Claire Monge, Jean-Yves Exposito, Danielle Campiol Arruda, Bernard Verrier

**Affiliations:** 1Laboratoire de Biologie Tissulaire et d’Ingénierie Thérapeutique, Institut de Biologie et Chimie des Protéines, UMR 5305, CNRS/Université Claude Bernard Lyon 1, 7 Passage du Vercors, CEDEX 07, 69367 Lyon, France; 2Precision NanoSystems Inc., 655 West Kent Avenue North Unit 50, Vancouver, BC V6P 6T7, Canada

**Keywords:** mRNA-vaccine, lipid nanoparticles, gag HIV-1, administration routes, heterologous protein boost, Th2/Th1 polarization

## Abstract

mRNA-based vaccines have made a leap forward since the SARS-CoV-2 pandemic and are currently used to develop anti-infectious therapies. If the selection of a delivery system and an optimized mRNA sequence are two key factors to reach in vivo efficacy, the optimal administration route for those vaccines remains unclear. We investigated the influence of lipid components and immunization route regarding the intensity and quality of humoral immune responses in mice. The immunogenicity of HIV-p55Gag encoded mRNA encapsulated into D-Lin-MC3-DMA or GenVoy-ionizable lipid-based LNPs was compared after intramuscular or subcutaneous routes. Three sequential mRNA vaccines were administrated followed by a heterologous boost composed of p24-HIV protein antigen. Despite equivalent IgG kinetic profiles of general humoral responses, IgG1/IgG2a ratio analysis showed a Th2/Th1 balance toward a Th1-biased cellular immune response when both LNPs were administrated via the intramuscular route. Surprisingly, a Th2-biased antibody immunity was observed when DLin-containing vaccine was injected subcutaneously. A protein-based vaccine boost appeared to reverse this balance to a cellular-biased response correlated to an increase in antibody avidity. Our finding suggests that the intrinsic adjuvant effect of ionizable lipids appears to be dependent on the delivery route used, which could be relevant to reach potent and long-lasting immunity after mRNA-based immunization.

## 1. Introduction

Messenger ribonucleic acid (mRNA)-based vaccine potency emerged in the vaccination field during the SARS-CoV-2 pandemic through the impressive success of two vaccines. Wolff and colleagues first proved that mRNA could be used as an active substance to induce an in vivo specific translation of a reporter gene [1]. For the next 30 years, clinical trials were extensively launched, demonstrating efficient antigen expression and immune response activation against animal and human infectious diseases [2]. This technology showed advantages to be produced and validated in a record time to immunize the worldwide population. This fast reactivity was possible due to pioneering works in the RNA-therapeutics field allowing the development of (1) nucleoside-modified mRNA to impede a strong inflammatory response, and (2) a lipid nanoparticle (LNP) delivery system [3,4]. It is only in 2020 that mRNA-based vaccines obtained their emergency use authorization [5]. During the past 3 years, mRNA-based therapeutics studies increased, reaching more than 700,000 publications overviewing several medical applications such as regenerative medicine, metabolic and genetic diseases, oncology, and infectious diseases [6].

mRNA nanocarriers are mainly represented by LNPs, which are composed of four components: an ionizable lipid (ALC-0315, SM-102, D-Lin-MC3-DMA, etc.), a structural phospholipid (usually DSPC), a cholesterol, and a polyethylene glycol (PEG) lipid [7]. The use of ionizable lipids containing amino groups with pKa < 7 is the main innovation that turned LNP technology into a suitable and efficient delivery system for mRNA. Due to their pH-sensitive properties, ionizable lipids are positively charged at acidic pH (pH < 6.0) and neutral or slightly charged at pH = 7.4 [8]. The complexation of mRNA and lipids is possible due to the self-assembly of components through electrostatic interaction and polarity modification, changing mixture hydrophobicity in the fluid stream using the microfluidic system to obtain mRNA-loaded LNPs. The downstream step enables decreasing the ethanol content in the post-chip production, as well as removing acidic residues by increasing acidic pH to physiological pH via buffer exchange. This step allows LNP stability by neutralizing positive toxic surface charges [9]. In this conformation, LNP is able to protect the highly sensitive RNA molecule against ubiquitous enzymes (ribonuclease) or physicochemical variations (pH, osmolarity, etc.). Then, after internalization of the LNP by the cells, the ionizable lipids recover their positive charges inside the endosomes and improve the release of the mRNA molecules into the cytosol via destabilization of the endosomal membrane [10]. In addition to the structural properties of each component, the ionizable lipid offers some intrinsic abilities that modulate the pharmacokinetics and pharmacodynamics of the LNP formulation [11,12,13]. The chemical structure of ionizable lipids can impact the LNP’s biodistribution [14,15], bioavailability [16], cellular tropism [12], and intrinsic immune adjuvant ability [17,18], which can modify short- and long-term in vivo efficacy.

After the COVID-19 mass vaccination campaign, several studies highlighted that mRNA-based vaccines were not able to induce long-term immunity [19,20]. The main approaches to improve in vivo efficacy are based on the selection and optimization of the RNA sequence [21] and the delivery system used [22]. Other improvements have been proposed such as heterologous vaccine regimens (consisting of an immunization scheme combining different types of vaccines [23]) or the change in administration route. Concerning mRNA-based vaccines, the intramuscular route is the most commonly used because it is easy to apply and possesses a high vascularization network facilitating the recruitment of the immune cells [11]. Other parenteral routes are largely known to improve immune response depending on the cellular population found at the injection site, such as the intradermal and intranodal routes [24]. Unfortunately, those administration sites are difficult to reach or can be invasive [25]. The subcutaneous delivery route remains an alternative for anti-infectious vaccines and has already demonstrated no differences, compared to the intramuscular route, after inadvertent administration of anti-COVID 19 mRNA vaccines into patients [26,27]. Although an efficient immune response was reached, the real biological benefit of this administration route remains unclear.

Such vaccine strategies were already applied for other infectious diseases including human immunodeficiency virus (HIV) in different animal models. However, despite being identified 40 years ago, no cure or preventive vaccine has been discovered, although several drug treatments permit viral load control and decrease the mortality of opportunistic life-threatening infections [28]. In fact, the multiple escaping mechanisms of this virus allow a long-term viral infection that destroys the immune system. Thus, major current aims to eradicate this virus consist of creating long-term and neutralizing immune responses able to target a broad range of HIV strains [29]. In this context, the use of an mRNA-based vaccine offers new perspectives, accelerating the HIV research field by facilitating the design of new potent antigens and empowering the cellular and antibody immune responses [30]. Indeed, mRNA prophylactic tools have already shown comparable polyfunctional antibody responses to proteins in nonhuman primates [31], and other studies have confirmed the ability to elicit tier 2 neutralizing antibodies able to reduce the risk of heterologous infections [32,33,34]. Those preclinical successes permitted the National Institute of Allergy and Infectious diseases to launch three experimental HIV vaccine clinical trials in 2022 (NCT05217641). Despite this new hope, HIV vaccine history has been punctuated by successive failures, reminding us to maintain our effort in the development and optimization of effective vaccines; comparing routes of mRNA delivery could be very helpful in designing new mRNA formulations.

In the current study, LNP-based vaccines were formulated with different ionizable lipids: D-Lin-MC3-DMA (DLin) and GenVoy^TM^ (GV), for the production of DLin- and GV-LNP, respectively. By comparing different formulation processes and through dynamic light scattering analysis, we emphasize the variation range of particle size and homogeneity of LNP suspensions. Then, after in vitro characterization of mRNA formulations encoding the type I HIV p55Gag polyprotein, we assess the immunogenicity of each formulation after intramuscular (IM) or subcutaneous (SC) injections. We monitor immune response kinetics and analyze the antibody avidity index. To test the influence of the immunization route and a potential correlation with lipid components, we analyze the immune polarization (Th2/Th1) before and after a heterologous protein boost, as this boost can exemplify the difference of the different priming schedule of mRNA vaccination.

## 2. Materials and Methods

### 2.1. Materials

DSPC (1,2-distearoyl-sn-glycero-3-phosphocholine) and PEG-lipid (1,2-dimyristoyl-rac-glycero-3-methoxypolyethylene glycol-2000) were purchased from Avanti Polar Lipids (Alabaster, AL, USA). D-Lin-MC3-DMA (DLin) ionizable cationic lipid was obtained from MedChemExpress (New Jersey, NJ, USA). Cholesterol, 3 M sodium acetate (pH 5.5), and tris-EDTA buffer were purchased from Sigma-Aldrich (L’Isle-d’Abeau, France). Dulbecco’s phosphate-buffered saline (1× DPBS), Dulbecco’s modified Eagle medium culture (DMEM) and fetal bovine serum (FBS) were purchased from Gibco (Dublin, Ireland). Phosphate-buffered saline (PBS), Tween-20, and urea were obtained from Euromedex (Souffelweyersheim, France). Absolute ethanol was purchased from Carlo Erba Reagents (Peypin, France). RIPA lysis buffer and 100× Halt Protease inhibitor cocktail were procured from Thermo Fisher Scientific (Waltham, MA, USA). β-Mercaptoethanol, 4× Laemmli sample buffer, and Precision Plus Protein^TM^ Dual Xtra Prestained Protein Standards were purchased from BioRad (Marnes-la-Coquette, France). A GenVoy-ILM^TM^ nonviral lipid nanoparticle kit was donated by Precision Nanosystems Inc (Vancouver, BC, Canada). mRNAs encoding for firefly luciferase (FLuc) and p55Gag HIV polyprotein were purchased from Trilink BioTechnologies (San Diego, CA, USA). The recombinant HIV-1 p24 antigen was purchased from Px Therapeutics (Grenoble, France). Polylactic acid nanoparticles (PLA-NP), named i-Particles^®^, were purchased from Adjuvatis (Lyon, France).

### 2.2. mRNA-Based Lipid Nanoparticle (LNP) Preparation and Characterization

The formulation of LNPs was processed using 130 μg of nonoptimized mRNA encoding p55Gag HIV or FLuc proteins diluted in sodium acetate buffer (pH 5, 100 mM) as the aqueous phase. The organic phase was composed of a lipidic mix of D-Lin-MC3-DMA or GenVoy-ILM^TM^ with the composition of ionizable lipids/DSPC/cholesterol/PEG-lipids at 50:10:38.5:1.5 or 50:10:37.5:2.5 molar ratios in absolute ethanol, respectively. Aqueous and organic solutions were prepared to obtain an N/P ratio of 7 or 8, which corresponds to the molar quantity of amine groups (N) from ionizable lipids and phosphate groups (P) from the nucleic acid backbone. Both solutions were loaded onto the IgniteNxGen cartridge using NanoAssemblr^®^ Ignite^TM^ apparatus and set up at an aqueous/organic phase flow rate ratio at 3:1, a total flow rate of 12 mL/min, and a temperature of 25 °C to allow mRNA encapsulation into LNPs. The LNP dispersion was subjected to buffer exchange with 1× DPBS (pH 7.2) and concentrated using an AMICON ultracentrifugal filter device (regenerated cellulose, 10 kDa NMWL, Millipore^®^ Merck), with this step representing the downstream process (DSP). The centrifuge was settled at 8 or 20 °C with a speed of 5000× *g*. The recovered LNP dispersion was stored at 4 °C before usage. Samples were diluted 1/10 in 0.22 μm filtrated 1× DPBS or 1 mM NaCl solution and characterized by dynamic light scattering (DLS) using a Zetasizer Nano ZS Plus (Malvern Instruments, Malvern, UK). DLS analyses were determined at 25 °C with a scattering angle of 173°, and zeta potential measurement was assessed using a Doppler velocimetry laser at a scattering angle of 12.5°. mRNA encapsulation efficiency was quantified using the Quant-IT^TM^ RiboGreen RNA Assay Kit (InvitroGen)**.**

### 2.3. Protein-Based Subunit Vaccine Formulation

HIV-1 p24 antigen (a subunit of p55Gag protein) was passively adsorbed onto PLA-NP, as previously described [35]. HIV-1 p24 was diluted in 1× DPBS at 400 μg/mL and mixed, V/V, with a PLA-NP suspension diluted at 1 mg/mL in 1× DPBS. The P24/PLA mix was incubated at room temperature, for 2 h under gentle agitation. The adsorption rate was calculated by dosing free p24 proteins present on supernatant obtained after two centrifugations (10 min at 10,000× *g*) of 200 μL of the p24/PLA complex. Proteins were dosed using the Pierce^TM^ BCA protein assay kit. Then, complexed formulations were diluted in 1× DPBS to reach 30 μg/mL of complexed p24. The formulation was diluted (1/10) in 0.22 μm filtrated 1 mM NaCl solution before DLS analysis.

### 2.4. In Vitro Luciferase Transfection Assay

The HeLa cell line (Invivogen, Toulouse, France) was maintained in a complete medium composed of DMEM culture supplemented with 10% heat-inactivated FBS. One day before transfection, 2 × 10^4^ cells/well were seeded on white 96-well plates (Greiner Bio-One, Courtaboeuf, France) with completed medium and incubated at 37 °C under 5% CO_2_. After 24 h, cells were transfected with LNP at 100 ng of encapsulated FLuc-mRNA/well in completed medium. TransIT^®^ transfection reagent (#MIR2225, Mirius) was used as a positive control to transfect FLuc mRNA, following the manufacturer’s instructions. Non-transfected cells were used as a negative control. The bioluminescence assay was performed, 24 h post-transfection, using the Bright-Glo^TM^ Luciferase assay system (Promega, Charbonnières-les-bains, France). Briefly, 100 µL of reconstituted reagent was added to each well and incubated for 5 min at room temperature. Luminescence intensity was measured on a Tecan i-control Infinite M1000 (Tecan, Männedorf, Switzerland).

### 2.5. In Vitro p55Gag Polyprotein Detection Assay

For p55Gag polyprotein characterization, HeLa cells were seeded in six-well plates with a density of 1 × 10^6^ cells/well 1 day prior transfection. DLin- and GV-based LNP containing p55Gag mRNA were diluted into 2.5 mL of complete medium (DMEM + 10% FBS), and 1 µg of encapsulated-mRNA was added per well. The TransIT^®^-mRNA Transfection Kit was also used as a positive control to transfect p55Gag mRNA and non-transfected cells were used as a negative control. Each condition was performed in duplicate.

To isolate intracellular proteins, cells were treated with a lysis solution composed of RIPA lysis buffer and Halt Protease inhibitor cocktail (1:100 (*v*/*v*), respectively). Briefly, 0.5 mL of lysis solution was applied per well on ice. Wells were scratched, and lysates were centrifuged at 7600× *g* for 10 min at 4 °C. Pellets were discarded, and supernatants were recovered for intracellular protein analysis. Protein concentrations were evaluated using the Pierce^TM^ BCA protein assay kit for Western blot analysis.

First, 20 µg of total protein was treated using reconstituted Laemmli/β-mercaptoethanol solution and heated at 95 °C for 10 min. Denatured samples were loaded onto precasted Mini-Protean TGX 4–15% precast gel (Biorad, Marnes-La-coquette, France), and electrophoresis was performed in sodium dodecyl sulfate (SDS) reducing conditions at 200 V for 30 min. After protein transfer on a nitrocellulose membrane using a TransTurbo-blot transfer system, membranes were saturated with 1× PBS/0.05% Tween 20/3% milk solution for 30 min. Then, anti-p24 HIV primary monoclonal antibody (#mmAbM01, purchased from Polynum Scientific GmbH, Klosterneuburg, Austria) was diluted at 1 µg/mL in 1× PBS/0.5% milk, added to the membrane for 1 h, and washed three times with 1× PBS/0.05% Tween-20 solution. Horseradish peroxidase (HRP)-conjugated goat anti-mouse immunoglobulin G (IgG) was put on a membrane for 1 h and revealed with Clarity^TM^ Western ECL substrate. A fusion camera was used to detect produced bioluminescence.

### 2.6. Immunization and Sampling

CB6F1 mice were purchased from Charles River Laboratories (L’Arbresle, France) and housed at the Plateau de Biologie Expérimentale de la Souris (PBES, ENS, Lyon, France). All procedures were approved by the Ethical Committee of Rhône-Alpes for the Animal Experimentation (CECCAPP, Lyon, France (authorization number ENS_2017_031)).

Four groups of mice were immunized during this study with freshly (<1 week) prepared formulations. Three sequential doses of mRNA vaccine were administrated into 7 week old CB6F1 mice (n = 8) at 4 week intervals. DLin- or GV-based LNPs were administrated via the intramuscular or subcutaneous routes using 3 µg of encapsulated p55Gag mRNA. Immunization schemes were completed with 3 µg of p24 HIV-1 protein loaded onto PLA-NP and administrated subcutaneously. Each subcutaneous administration was made in the axillar area (100 µL), while intramuscular delivery was realized on quadriceps muscles (50 µL per quadriceps), both with a 30G/1/2″ Omnican^®^ 50 U-100 insulin syringe (Braun).

At different times of the vaccinal schedule, approximately 100 µL of blood samples were collected from the orbital sinus under 4% isoflurane sedation. Blood was heated at 37 °C for 30 min and centrifuged twice for 10 min at 10,000× g. Collected serum was stored at −20 °C and used for ELISA.

### 2.7. Anti-p24 Antibody ELISA

Nonsterile 96-well plates (ThermoFischer, Waltham, MA, USA) were coated with 1 µg/mL of p24 HIV protein in 1× DPBS overnight at room temperature. Plates were blocked by 10% milk in 1× DPBS for 1 h and washed three times with 1× PBS/0.05% Tween-20 washing buffer. A serial dilution of mouse serum samples in 1× DPBS/1% BSA was added to the wells and incubated for 1 h at 37 °C; then, the wells were washed three times with the washing buffer. Horseradish peroxidase (HRP)-coupled goat anti-mouse antibodies (IgG, IgG1, and IgG2A, all purchased from Southern Biotech) were diluted at 0.1 µg/mL in 1× DPBS/1% BSA, and added to the wells. After 1 h at 37 °C, wells were washed three times with the washing buffer. Lastly, antibodies coupled to HRP were revealed using 100 µL of reconstituted TMB substrate reagent (BD Bioscience, #555214), and reactions were stopped 30 min later with 1 N sulfuric acid (VWR, #32053.602). The adsorption was read at 450 nm, and optical density (OD) was corrected with adsorption at 630 nm using a Multiskan FC plate reader (Thermo Scientific, Waltham, MA, USA) [35]. Antibody titers corresponded to the higher sample dilution, leading to an OD above the predetermined cutoff at day 0. The avidity assay was performed following the same protocol than anti-p24 ELISA assay, except for the washing step before incubation of conjugated antibodies. During this step, plates were washed three times with 8 M urea diluted into 1× PBS/0.05% Tween-20. The avidity index (%) represents the percentage of antibodies binding antigens after urea treatment compared to ELISA without urea treatment [36].

### 2.8. Statistical Analysis

All statistical analyses were performed using GraphPad Prism version 8.0 software (San Diego, CA, USA). All data were collected and expressed as the mean ± standard deviation (SD). Lipid nanoparticle DLS characteristics were analyzed with a multiple unpaired t-test, the luciferase transfection assay was analyzed using a two-way ANOVA parametric test, the humoral immune response monitoring, before and after protein heterologous boost, was analyzed using the Wilcoxon or Mann–Whitney U nonparametric test; and correlation analysis was performed using the Pearson or Spearman test, for parametric and nonparametric data, respectively.

## 3. Results

### 3.1. Effect of the N/P Ratio and the Purification Temperature on Colloidal Physicochemical Characteristics of mRNA-LNP Formulations in Different Dispersants

LNPs were formulated with FLuc mRNA at N/P ratios (NP) of 7 and 8. Those LNPs differed mainly by the ionizable lipid used and slightly by their molar ratios of cholesterol to PEG-lipid (38.5:1.5 for DLin-LNP and 37.5:2.5 for GV-LNP). The impact of DSP (buffer exchange) temperature and the dispersing agent used for DLS analysis on LNP physicochemical characterization was assessed.

After the self-assembly process, residual ethanol was removed during the DSP by centrifugation at 8 and 20 °C (designated as DSP 8 and DSP 20), and DLS analysis was performed in 1× DPBS (Table 1). All LNPs presented a hydrodynamic diameter (Hd) between 70 and 90 nm and were homogeneous in size with a polydispersity index (PI) below 0.15. Even so, both formulations appeared to have an opposite behavior under a high-temperature DSP. A significant but slight increase in Hd and PI were observed when DLin-LNPs were purified at DSP 20. Zeta potential analysis and mRNA encapsulation efficiency showed no differences, maintaining low negative surface charges and a high mRNA loading rate (>90%).

The same formulations were used to compare colloidal fluctuations after LNP dilution into 1 mM NaCl before DLS analysis (method used for polymeric nanoparticles [37]) (Table 2). A hypotonic environment significantly increased all particle sizes, especially those with GV-LNPs. More precisely, the Hd of GV-LNPs significantly increased by 30–40 nm when the osmolarity and pH decreased. Regarding DLin-LNPs, osmotic variation had less impact on Hd with +11/+6 nm for NP7/8, allowing nanoparticles with sizes below 95 nm, along with a slight increase in solution heterogeneity only at low NP (PI of 0.089 to 0.12 at DSP 8). Surface charges significantly increased after NaCl dilution, reaching a maximum of +5.4 mV, except for DLin-based formulations at NP 8.

For the remainder of this study, FLuc and p55Gag mRNAs were encapsulated into DLin- and GV-based LNPs using parameters producing a lower particle size and polydispersity variability between them. Considering that potent in vivo efficacy was demonstrated when LNP formulations had a mean size <200 nm and a PI <0.2 [38,39], LNPs were formulated using an NP ratio of 7 and a DSP temperature of 20 °C.

### 3.2. Both mRNA-LNPs Showed Efficient Cell Transfection and Translation In Vitro

To confirm if LNPs are able to correctly deliver mRNA and facilitate protein expression, HeLa cells were transfected with formulations carrying a reporter gene encoding FLuc. The luciferase assay revealed an efficient expression of Fluc protein for all tested conditions, with the density of cells analyzed being similar (Figure 1a). Nevertheless, DLin-LNPs showed significantly lower expression of Fluc compared to GV-LNPs and TransIT^®^ positive control.

With the aim of validating these mRNA complexes for in vivo application, mRNA encoding p55Gag HIV protein was used instead of Fluc mRNA. This polyprotein Gag (55 kDa) is a precursor of several proteins responsible for recruiting viral components to the cell membrane and forming the virus core structure. Furthermore, an immune response specific to anti-p24 antigen (a subunit of p55 at 24 kDa) was associated with viral load control in people living with HIV, making it an interesting model for our in vivo immunization assay [40,41]. The production of the p55Gag protein in HeLa cells was detected using an anti-p24 monoclonal antibody by Western blot, 48 h after LNP/p55Gag mRNA transfection (Figure 1b). For both LNPs and the TransIT^®^ positive control, the immunoblot revealed an immunoreactive band around 50 kDa in the cellular compartment, corresponding to the size of the p55Gag polyprotein.

### 3.3. Administration Route of p55Gag mRNA-Based Vaccine Does Not Impact the Anti-p24 Humoral Immune Response Kinetics and Antibody Avidities

On the basis of the in vitro evidence of p55Gag expression, the influence of the ionizable lipids used in LNP formulations on immune responses were compared following two different routes of administration in mice. CB6F1 mice were immunized three times at 4 week intervals with 3 µg of encapsulated p55Gag mRNA (Figure 2a). Each LNP was administrated subcutaneously (SC) and compared to the gold standard represented by the intramuscular (IM) route.

Total IgG responses were monitored after each injection (at weeks +2/+6/+10) to evaluate the prime immune response and boost effects after the second and third shots (Figure 2b,c). Two weeks after the first injection, a more homogeneous immune induction for DLin-LNP-immunized mice was observed, while GV-LNP groups showed no or low responses in mice (5/8 mice for SC and 2/8 for IM). The second shot was able to significantly increase IgG titers (>10^5^) corresponding to a typical vaccine boost effect. In contrast, the third dose did not enhance immune response intensity (Figure 2b,c). No significant differences were found when both formulations were compared (*p* > 0.05).

In order to estimate the quality of the humoral immune response, the serum avidity indices of mice immunized with DLin-LNPs (Figure 2d) and GV-LNPs (Figure 2e) were estimated. Below 30%, antibodies are considered to have weak avidity, with a serum avidity above 50% indicating strong avidity, and values between denoting intermediate avidity. The third administration of DLin-LNPs did not impact antibody quality, with the avidity indices ranging mainly in the weak avidity zone for SC administration and the weak-to-intermediate zone for IM administration (Figure 2d). In the case of GV-LNP SC (Figure 2e), the immune response seemed to induce more heterogeneous avidity, ranging from weak to high avidity indices compared to the IM route, which mainly revealed mice secreting anti-p24 antibodies with intermediate avidities. In either case, humoral response intensity and quality did not show significant differences between the LNP vaccine used and the delivery route applied during the mRNA immunization protocol.

To exemplify a putative difference in mRNA priming efficacy according to the nature of lipids or the administration route, we performed a protein boost using the HIV p24 antigen loaded onto PLA-NP [42]. The titer and quality of secreted anti-p24 antibodies were measured as previously described after mRNA injection, at weeks +2 and +11 after protein boost (Figure 3). In all groups, the subcutaneous protein boost maintained total IgG titers above 10^5^ for 11 weeks post injection. After the heterologous boost, no significant difference was observed according to administration route and ionizable lipid used (Figure 3b,c), suggesting that a plateau level was reached, by each schedule of immunization. In a similar manner, this protein boost did not improve immune response quality according to avidity index at each timepoint (Figure 3d,e), except for DLin-LNP SC, where the antibody avidity shifted from the weak zone to the intermediate-to-high zone after protein boost (Figure 3d).

### 3.4. Immune Polarization Mediated by IgG1/2a Ratio Shows Contrasting Effect in an Ionizable Lipid- and Administration Route-Dependent Manner

The immune response polarization of the different vaccine assays (LNP used, route of administration) was studied during the immunization regimen shift, before and after the heterologous protein boost. IgG1 and IgG2a subtypes were titrated to calculate the IgG1/2a ratio from weeks +10 and +14 in mice sera (Figure 4a). A ratio below or above 1 is representative of a cellular T-helper cell 1 (Th1) or an antibody Th2-biased immune response, respectively. Additionally, antibody avidities were compared before and after the protein boost (Figure 4b).

Before the protein boost, GV-LNPs tended to promote cellular Th1-biased immune response as judged by an IgG1/IgG2a ratio mean of 0.1 and 0.01 for SC and IM mice groups, respectively. After p24 protein boost, no significant difference in polarization was observed for both mRNA-GV-LNP immunization delivery routes (Figure 4a). The protein boost in GV-LNP-treated mice did not significantly improve the antibody quality (Figure 4b).

In contrast, DLin-LNP-treated mice exhibited different behavior depending on the administration route used. Following the complete mRNA immunization scheme via the SC delivery route, this formulation tended to induce an antibody Th2-biased immune response (IgG1/2a ratio around 2) in comparison to IM injection, which promoted a cellular Th1-biased immune response (IgG1/2a ratio around 0.025). The P24-protein boost shot promoted a significant inversion of the mean IgG1/2a ratio when DLin-LNP was initially injected via the SC route (Figure 4a), leading to a cellular Th1-biased immune response. In this mouse group, a significant improvement in antibody quality was observed during the immunization regimen shift, from a mean of 26.9% ± 7.4% to 41.2% ± 12.4% at weeks +10 and +14, respectively (Figure 4b).

### 3.5. DLin-LNP SC and Heterologous Boost Combination

Previous results of this study suggested a relation between the ionizable lipid used during the LNP self-assembly process and the delivery route applied, in combination with a protein boost. To highlight the positive impact of the protein boost regarding immune polarization, IgG1/2a ratios and avidity indices from weeks +10 and +14 were used to determine a potential correlation (Figure 5) between Th2/Th1-biased induced immune responses and antibody qualities.

When DLin-LNPs were initially injected (Figure 5a,b), before the protein boost, a significant correlation was observed between antibody quality and a switch from Th2 to Th1 polarization (Figure 5a,b). More interestingly, this analysis suggested an opposite correlation depending on administration route used. When DLin-LNP was injected via the SC route, a positive correlation was found (r = 0.5255 and *p* = 0.0385), while a negative correlation (r = −0.5376 and *p* = 0.0317) was observed after IM injection. In parallel, alternative administration routes for GV-LNP seemed not to be involved in a positive or negative progression of humoral immune response (r = −0.37 and *p* = 0.1584 for SC route and r = −0.05383 and *p* = 0.8430 for IM route, Figure 5c,d). From these data, we could observe that the protein boost after mRNA priming DLin-LNPs injected via the SC route was able to promote the quality of the antibody response.

## 4. Discussion

mRNA-based vaccines have already shown several benefits to efficiently protect against infectious diseases in humans. These achievements were reached through successive and iterative progress, both in mRNA delivery through LNP vehicles and in mRNA design. It has been widely documented that mRNA vaccine technology is easy- and fast-to-produce, cost-effective, and highly versatile, offering new perspectives to treat a large number of pathologies [43,44,45]. As global warming offers more suitable conditions to improve pathogen dissemination, human adaptation, and epidemic outbreak, mRNA vaccines will become a unique tool to propose a fast preventive deployment strategy onsite to interfere with new fast-adapting and -spreading human pathogens [46,47]. Thus, understanding precise mechanisms controlling the colloidal behaviors of LNP dispersions could permit elaborating accurate quality controls or tools to analyze LNP batches after synthesis and before in vitro validation.

LNPs are usually prepared through a three-step process: the self-assembly of components, the downstream process, and the in vitro control quality process. Each step is designed to confer and maintain electrostatic stability between the carried mRNA and the delivery systems (supported by the ionizable lipid and stabilized by weak interaction due to structural lipids). In this work, we formulated LNPs using a downstream process temperature settled at 8 °C and 20 °C (Table 1). DLS analysis (using conventional 1× DPBS dispersing agent) showed that both LNPs were significantly impacted by an increased DSP temperature when formulated at low NP. Furthermore, when the same formulations were diluted into 1 mM NaCl to create a hypotonic environment with a slight pH decrease, GV-LNPs showed higher Hd and PI variation than DLin-LNP, suggesting different behaviors of those formulations depending on the suspension environment.

The main difference between the two types of LNPs was the ratio of cholesterol to PEG-lipid (38.5:1.5 for DLin-LNP and 37.5:2.5 for GV-LNP). It has been reported that cholesterol contributes to the LNP structural integrity modulating the membrane fluidity/rigidity, and that PEG-lipid protects the particle shell, minimizing LNP agglomeration [48,49]. Consistent with these observations, the higher content of cholesterol in DLin-LNPs could allow better size stability compared to GV-LNPs, showing a significant increase in Hd (Table 2). The improved stability could be due to a decreased extra/intra-vesicular osmotic flow caused by a lower rate of cholesterol. Moreover, Kulkarni et al. already suggested that cholesterol improved membrane rigidity, conferring reduced drug leakage from lipid particles [10]. These parameters could directly impact in vivo processing by modifying the half-life of LNPs, as well as their ability to pass through lymphatic vessels and induce a potent immune response directly into lymph nodes [43,50]. Moreover, the choice of ionizable lipids could impact the efficiency of mRNA release into the cytosol. This difference might have contributed to the higher luciferase expression obtained in HeLa cells transfected by GV-LNP compared to DLin-LNP (Figure 1a), but could not be correlated to the in vivo efficacy, as reported in [51]. Despite the increased sizes and PI (<200 nm and PI < 0.2; Table 2), the formulations in our study showed efficient transfection and translation abilities in the HeLa cell line (Figure 1), validating our LNPs for in vivo applications to induce a specific immune response.

In order to emphasize the in vivo benefits of LNPs combined with an alternative parenteral delivery route, we compared the SC administration to the IM route currently used in human mRNA vaccine injections [11,25]. We encapsulated the p55Gag mRNA sequence coding for p24 subunit antigen, which is used as a vaccine immunogen because of its anti-p24 response associated with viral control in people living with HIV [40]. The initial mRNA immunization protocol consisted of a homologous prime/boost/boost scheme with both formulations. The interval between injections was selected according to Garcia-Dominguez et al.’s report suggesting that an increased interval impacts the robustness and durability of specific SARS-CoV-2 immune responses. Independently of the delivery route, two immunizations were sufficient to induce high and homogeneous anti-p24 IgG responses. However, the third injection did not improve total IgG titers or avidity indices (Figure 2). Such similarity between injection sites and immune response kinetics is in agreement with clinical data after inadvertent administration of SARS-CoV-2 mRNA vaccine [26]. Surprisingly, IgG1/2a ratios from week +12 sera revealed a significant difference concerning Th2/Th1 adaptative immunity bias.

Independently of the ionizable lipid used to formulate LNP, intramuscularly immunized mice induced a Th1-biased polarization (Figure 4a). This cellular-biased immune response seemed to have a similar effect to the BNT162b2 SARS-CoV-2 mRNA vaccine via the same delivery route. In this case, ALC-0315 ionizable lipid was used to formulate LNPs and appeared to induce a cellular-biased innate immunity promoted by the secretion of interleukin-6, interleukin-15, and gamma-interferon [52,53]. Interestingly, Valentin et al. reported a strong antibody but not cellular immune response on immunized nonhuman primates when a specific ionizable lipid (property of Acuitas Therapeutics) was used to formulate p55Gag mRNA into LNPs before injection via the IM route. They observed that this Th2-biased polarization was promoted only at low dose of administrated mRNA [54]. These data suggest that the ionizable lipid-containing LNPs might have an intrinsic adjuvant effect with some evidence of a dose–response relationship. Moreover, we showed in this study that DLin-LNPs induced antibody Th2-biased immunity when subcutaneously administrated (Figure 4a). These data support the hypothesis that LNPs inducing a Th2- or Th1-biased adjuvant activity could be dependent on the ionizable lipid used, as well as the delivery route applied.

Furthermore, the mRNA immunization protocol was completed with a single protein boost composed of p24 antigen complexed with PLA-NP, before being administrated subcutaneously (Figure 3a). This heterologous boost was able to maintain high anti-p24 IgG titers and intermediate avidity indices for 11 weeks until the end of the study (Figure 3). Despite that, this injection promoted a Th1-biased response significantly altering the immune polarization of the DLin-LNP SC-vaccinated mice (Figure 4a). More precisely, this Th2/Th1 inversion represented a significant improvement in antibody quality (Figure 4b) positively correlated to the SC administration route (Figure 5a). On the contrary, DLin-LNPs administrated via the IM route showed a negative correlation between the Th2/Th1 balance and avidity evolution, before and after heterologous boost (Figure 5b). Concerning GV-LNP, the p24 protein boost maintained low IgG1/2a ratios (Th1), which presented no significant impact on antibody quality and the delivery route (Figure 4a,b and Figure 5c,d). The opposite effects observed between the type of LNP and the administration route indicate that switching the Th2 to Th1 balance improved the antibody quality compared to a sustained Th1-biased immune response. On the basis of these results, it will be interesting to carry out this subcutaneous immunization regimen (DLin-LNP, subunit vaccine), but choosing a more relevant HIV antigen such as the Envelope glycoprotein. Guiding the immune polarization during a heterologous vaccination protocol could be of great interest for highly diversifying pathogens, such as HIV, requiring antibodies and T-cell responses to control the infection [28]. Both are involved in inducing somatic hypermutation in germinal center B cells responsible for long-term broadly neutralizing antibodies [17,55].

## 5. Conclusions

In this study, we showed that DLin- and GV-containing LNPs could be formulated using a downstream process temperature at 20 °C. Then, LNPs were subcutaneously injected and compare to the gold-standard IM route. The delivery system and the electrostatic stability of the obtained RNA-loaded LNPs were two key factors influencing in vivo efficacy. Mouse immunization experiments showed equivalent IgG titers and avidity indices during the mRNA immunization protocol. Nonetheless, we demonstrated that efficacy could be dependent on the ionizable lipid and administration route used, leading to an antibody Th2-biased immune response when DLin-based LNPs were subcutaneously injected. This polarization could be modified by a heterologous boost, which was positively correlated to an antibody quality improvement in the same animal group. We hypothesize that integrating the delivery route and the ionizable lipid selection during specific mRNA vaccine development could enhance both Th1 and Th2 polarization, which can affect the efficiency of the immune response when the immunization protocol is combined with a heterologous boost.

## Figures and Tables

**Figure 1 pharmaceutics-15-01009-f001:**
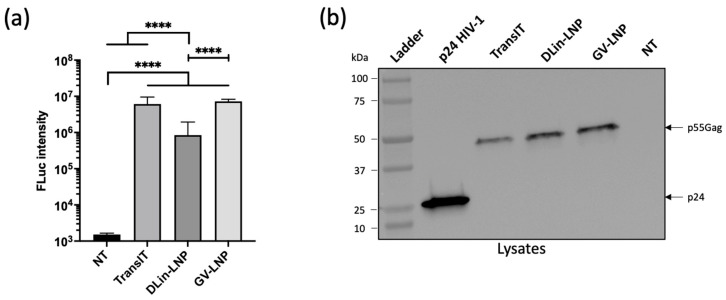
LNP-Fluc or p55Gag mRNAs purified at 20 °C show efficient in vitro transfection and mRNA translation. (**a**) Untreated HeLa cells (NT), as well as Fluc mRNA (TransIT) and Fluc mRNA/LNP nanocomplexes (DLin-LNPs and GV-LNPs), were transfected in complete medium (100 ng/well). The luciferase activity was measured 24 h post transfection. (**b**) p24 recombinant protein and total cellular proteins (20 µg per well) from HeLa cells transfected with TransIT^®^ or LNP-p55Gag mRNA complexes were separated onto a gradient SDS/PAGE gel (4–15%) and analyzed by Western blot using an anti-p24 monoclonal antibody specific of the HIV Gag protein. Luciferase intensity data are presented as the mean ± SD; two-way ANOVA statistical analysis was applied, followed by a Dunnett’s multiple comparison test; **** *p* < 0.0001.

**Figure 2 pharmaceutics-15-01009-f002:**
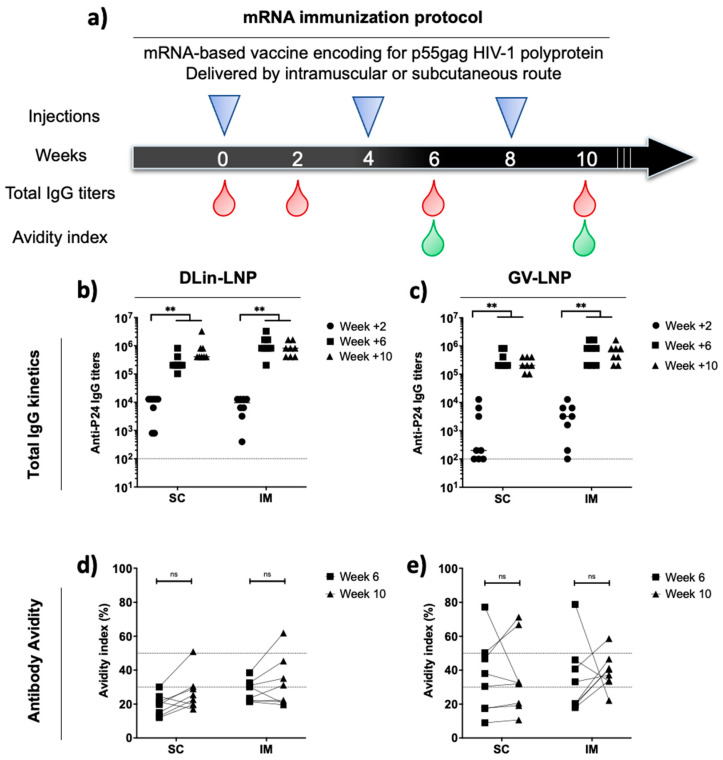
p55Gag mRNA-LNP-based vaccines elicit equivalent IgG titer kinetics with a higher heterogeneity of antibody avidity index for GV-LNP. (**a**) The vaccination schedule: CB6F1 mice received three sequential shots (4 week intervals) of DLin- or GV-LNPs carrying 3 µg of mRNA encoding p55Gag polyprotein. Blood samples were collected at different intervals in order to evaluate IgG titers (**b**,**c**) and antibody avidity indices (**d**,**e**) of mice vaccinated with DLin-LNPs (**b**,**d**) and GV-LNPs (**c**,**e**). Each point represents an individual mouse. The nonparametric Wilcoxon test was applied to highlight significant differences (*p* > 0.05: NS and *p* < 0.01 **).

**Figure 3 pharmaceutics-15-01009-f003:**
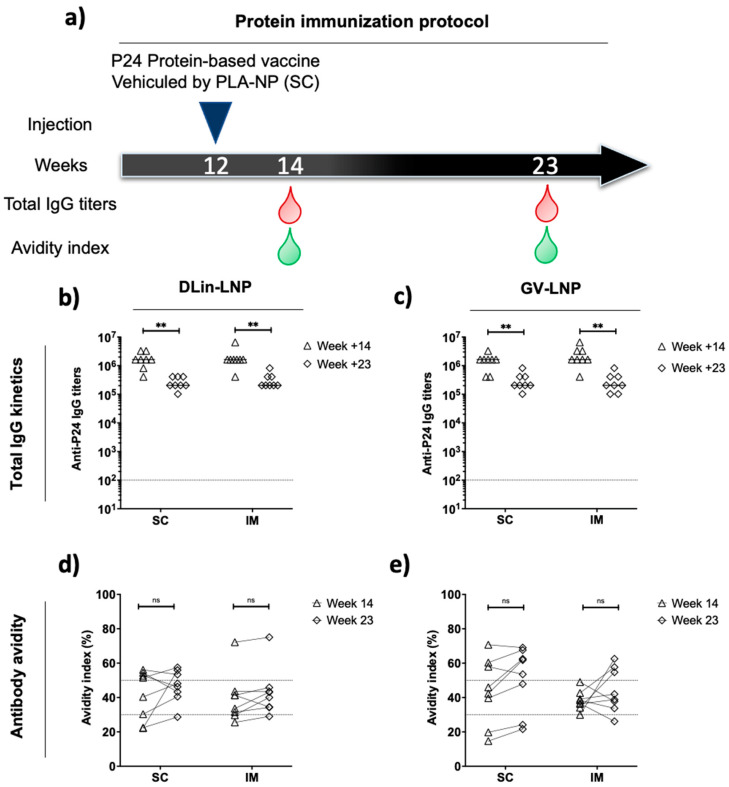
Subcutaneous protein heterologous boost maintained high IgG titers and did not interfere with antibody quality. Four weeks after the last mRNA injection (**a**), mice received protein-based vaccine composed of p24Gag protein loaded onto PLA-NP. Then, 3 µg of protein was administrated subcutaneously to evaluate humoral immune response (**b**,**c**) and its avidity (**d**,**e**). The nonparametric Wilcoxon test was assessed to determinate significant differences (*p* > 0.05: NS and *p* < 0.01 **).

**Figure 4 pharmaceutics-15-01009-f004:**
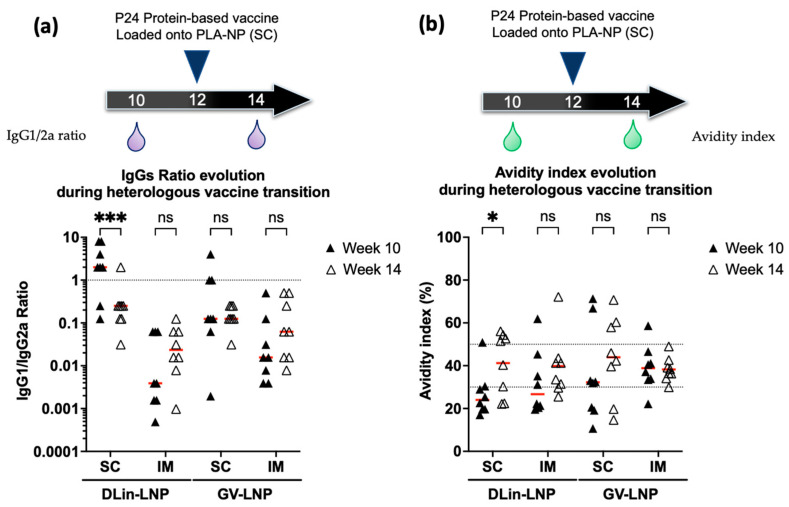
Protein boost in SC injected mice with DLin-LNP significantly changed immune response polarization and antibody quality. Specific IgG1/IgG2a ratios (**a**) and avidity indices (**b**) were measured before and after heterologous protein boost. The dotted line in (**a**) represents the equivalent cellular/humoral response. The dotted lines in (**b**) underline the 30% and 50% avidity index borders. Below 30%, antibodies are considered to have weak avidity, whereas values above 50% indicate strong avidity, and those between denote intermediate avidity. The red lines represent the means of each group. The Mann–Whitney U nonparametric test was used to reveal significant differences (*p* > 0.05: NS, *p* < 0.05 *, and *p* < 0.001 ***).

**Figure 5 pharmaceutics-15-01009-f005:**
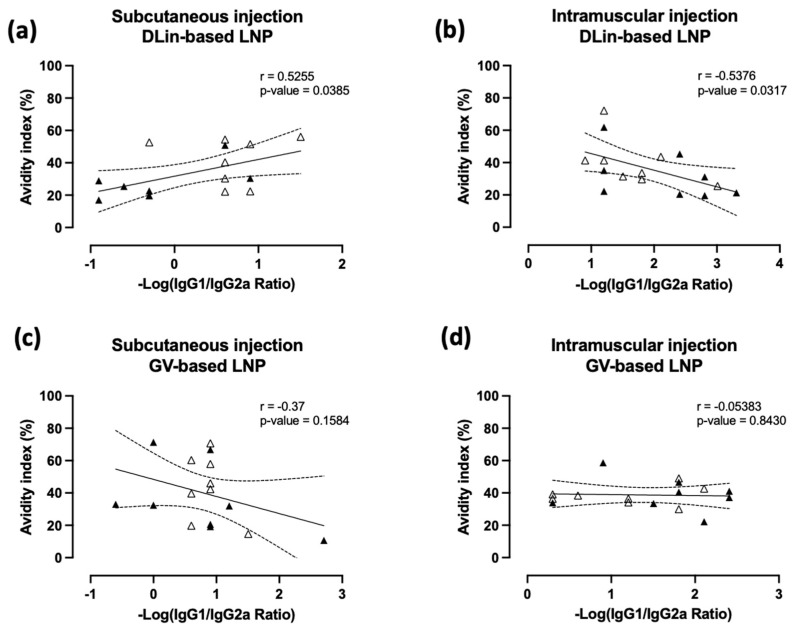
DLin-based LNPs showed opposite effects in a delivery route-dependent manner. The correlation between IgG1/IgG2a ratios and avidity indices obtained in DLin-LNP (**a**,**b**) and GV-LNP (**c**,**d**) treated mice after SC (**a**,**c**) and IM (**b**,**d**) injections. Data from weeks +10/+14 were used to determinate relations between IgG1/IgG2a ratios and avidity indices during mRNA protein immunization schedule. Each data point represents an individual animal before (black triangle) and after (white triangle) protein boost. For each formulation and delivery route, statistical analysis was performed to construct the correlation curve and 95% confidence index (continuous and dotted lines, respectively). The Spearman or Pearson correlation test was applied to determine the linear correlation coefficient (r) and its associated *p*-value (*p* < 0.05: not significant; *p* > 0.05: significant).

**Table 1 pharmaceutics-15-01009-t001:** Characterization of DLin- and GV-based LNP/FLuc-mRNA into DPBS after DSP temperature variation.

LNP-Based Vaccine	N/P Ratio	DSP Temperature	Hydrodynamic Diameter (nm) ^(1)^	Polydispersity Index (PI) ^(1)^	Zeta Potential (Zp)	mRNA Encapsulation Efficiency (%)
DLin-LNP	7	8 °C	75 ± 0.7	0.089 ± 0.015	−3.7 ± 0.5	95.4 ± 0.8
	8		68 ± 2	0.068 ± 0.016	−6.6 ± 1.6	90.2 ± 1.4
	7	20 °C	88 ± 3 ****	0.139 ± 0.007 **	−5.2 ± 1.4	92.8 ± 1.7
	8		73 ± 0.9 *	0.087 ± 0.004	−6.5 ± 1.1	91.9 ± 1.8
GV-LNP	7	8 °C	90 ± 3	0.142 ± 0.002	−3.3 ± 1.0	95.4 ± 0.8
	8		75.6 ± 1.5	0.085 ± 0.021	−5.3 ± 0.2	90.3 ± 0.3
	7	20 °C	78 ± 2.8 ****	0.130 ± 0.012	−3.7 ± 0.4	92.8 ± 1.7
	8		74 ± 0.4	0.086 ± 0.009	−4.4 ± 0.1	94.3 ± 1.8

^(1)^ Statistical analysis was performed through multiple unpaired *t*-tests using formulations at DSP 8 °C as a reference (*p* < 0.05 *, *p* < 0.01 **, and *p* < 0.0001 ****). Data are presented as the mean ± SD. Experiments were conducted in triplicate.

**Table 2 pharmaceutics-15-01009-t002:** Characterization of DLin- and GV-based LNPs under hypotonic condition.

LNP-Based Vaccine	N/P Ratio	DSP Temperature	Hydrodynamic Diameter (nm) ^(1)^	Polydispersity Index (PI) ^(1)^	Zeta Potential (Zp) ^(1)^
DLin-LNP	7	8 °C	86.5 ± 3.4 **	0.12 ± 0.051	2.6 ± 0.1 ****
	8		74.2 ± 1.7 *	0.06 ± 0.003	−6.6 ± 0.3
	7	20 °C	93.7 ± 3.8 *	0.140 ± 0.069	3.1 ± 0.3 ****
	8		81.1 ± 1.7 **	0.074 ± 0.010	−5.7 ± 0.7
GV-LNP	7	8 °C	128.1 ± 4.4 ****	0.15 ± 0.013	4.4 ± 2.3 ****
	8		108.9 ± 2.64 ****	0.19 ± 0.020 ****	3.1 ± 1.7 ****
	7	20 °C	110.1 ± 8.1 ****	0.164 ± 0.031 *	5.4 ± 1 ****
	8		115.2 ± 2.2 ****	0.162 ± 0.006 ***	0.3 ± 0.5 ****

^(1)^ Statistical analysis was assessed through multiple unpaired *t*-tests using 1× DPBS DLS results as a reference (*p* < 0.05 *, *p* < 0.01 **, *p* < 0.001 ***, and *p* < 0.0001 ****). Data are presented as the mean ± SD.

## Data Availability

Not applicable.

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
