# Peer review of "DLin-MC3-Containing mRNA Lipid Nanoparticles Induce an Antibody Th2-Biased Immune Response Polarization in a Delivery Route-Dependent Manner in Mice"

_pharmaceutics, 2023, doi:10.3390/pharmaceutics15031009_

Round 1
Reviewer 1 Report
This manuscript investigated the influence of specific lipid nanoparticle's components (i.e., DLin- and GVcontaining LNPs). The authors observed a Th2-biased antibody immune response when mRNA vaccines with DLin-based LNP are injected subcutaneously. This study suggests that the adjuvant effect of ionizable lipids may be dependent on the delivery route.
The manuscript is clear and of interest in mRNA vaccine field. Some minor issues need to be fixed before considering publication:
1. The alignment of caption in Fig. 2-5 is wrong.
2. In the introduction lines 43-44: why using "1/" and "2/" as the numbering of two points? It is common to use "(1)" and "(2)".
3. In Fig. 1a, DLin-LNP showed significantly lower Fluc Internsity than both GV-LNP and TransIT, which does not fully support the statement of “LNPs are able to protect mRNA and facilitate their uptake into cells”. It is expected to have more discussions on this.
4. Also, there is no directly comparison between DLin-LNP and GV-LNP in the following in vivo immune response experiment; the comparisons focus on SC vs. IM and week 6 vs. week 10. It is of interest to see if the lower in vitro translation level of DLin-LNP results in lower in vivo immune response compared to GV-LNP.
Reviewer 2 Report
pharmaceutics-2259068
DLin-MC3 containing lipid nanoparticles induce an antibody Th2-biased immune response polarization in a delivery route dependent manner in mice
The manuscript by Yavuz et al. described the preparation of lipid NPs loaded with HIV-p55Gag encoded mRNA as a vaccine. The authors compared two LNP formulations with two different ionizable lipids as well as two administration routes. Overall, the data are sufficient to support the conclusion. However, the authors should consider some comments below to improve the manuscript.
1. Introduction: The authors discussed different aspects of the mRNA-LNP vaccine. However, the part relating to HIV is superficial and needs to be expanded. The authors should clarify the status of the HIV vaccine, the research gap, as well as the novelty and significance of this study.
2. mRNA-based lipid nanoparticle (LNP) preparation and characterization (lines 123 – 143): The authors should clarify the amount of mRNA and which step was for mRNA encapsulation.
3. Methods: The authors should cite references for the methods used where relevant.
4. Immunization study on mice: The authors should describe how many groups were used.
5. Results and discussion: some information about the methods was mentioned or repeated in the Results and discussion parts. The authors should check and re-organize them.
6. The authors should clarify the basis for choosing the 2nd and 3rd doses at 4 and 8 weeks after the 1st dose.
7. How about the stability of LNPs and entrapment efficiency of the mRNA during storage?
8. Table 2: The authors should correct the decimal separators.
9. Based on the results, what is the optimal formulation/ combination for further studies?
Round 2
Reviewer 2 Report
The manuscript can be accepted as is.